# Learning Rich Nearest Neighbor Representations from Self-supervised Ensembles

## Abstract

Pretraining convolutional neural networks via self-supervision, and applying them in transfer learning, is an incredibly fast-growing field that is rapidly and iteratively improving performance across practically all image domains. Meanwhile, model ensembling is one of the most universally applicable techniques in supervised learning literature and practice, offering a simple solution to reliably improve performance. But how to optimally combine self-supervised models to maximize representation quality has largely remained unaddressed. In this work, we provide a framework to perform self-supervised model ensembling via a novel method of learning representations directly through gradient descent at inference time. This technique improves representation quality, as measured by k-nearest neighbors, both on the in-domain dataset and in the transfer setting, with models transferable from the former setting to the latter. Additionally, this direct learning of feature through backpropagation improves representations from even a single model, echoing the improvements found in self-distillation.

## 1 Introduction

The widespread application of pretrained convolutional neural networks in computer vision is one of the most important tools in the field. State-of-the-art on many benchmarks ranging from classification, to object detection, to pose estimation has been set using a pretrained model, such as an ImageNet classifier, as a network initialization (Kornblith et al., 2019; He et al., 2019; Chen et al., 2020a; Grill et al., 2020; Kolesnikov et al., 2019). Transfer learning is an entire field focused on studying and utilizing this phenomenon. While supervised ImageNet classifiers were the dominant feature extractors of choice for many years, recently self-supervised models have begun to take their place. Methods such as MoCo(v2), SimCLR(v2), SimSiam, PIRL, Swav, BYOL, and Barlow Twins all claim transferability competitive with or *superior to* that of ImageNet classifiers (He et al., 2019; Chen et al., 2020c;a;b; Chen & He, 2020; Misra & Maaten, 2020; Caron et al., 2021; Grill et al., 2020; Zbontar et al., 2021). As such, the question of what initialization to use has arisen; benchmark studies have sought to compare methods under dozens of different settings (Goyal et al., 2019; Zhai et al., 2019). Even when a decision has been made to use a particular feature extractor, the utility and knowledge of other options is then left unutilized.

To address this concern, we consider ensembling, a common practice in the supervised setting (Dietterich, 2000; Hinton et al., 2015). Ensembling models involves combining the predictions obtained by multiple different models in some way, typically by averaging or a similar operation. In the supervised setting, such outputs are aligned and such an operation easily captures the combined knowledge of the models. In the self-supervised setting, however, such alignment is not guaranteed, particularly when dealing with independently trained contrastive learners which many pretrained models of choice are. Averaging the features still is useful, and obtains reasonably strong image representations (Section 4), but we show that it is possible to build an ensembling strategy that yields richer, stronger representations than the mean feature. We do so without training any new CNN components, allowing for the same frozen backbone to be used across applications.

How to approach ensembling in the self-supervised setting? We contend that *the goal of a model ensemble is to capture the useful information provided by the different models in a single representation*. We consider the "capture" of information from a recoverability perspective: if every network's features can be recovered by some fixed operation on a representation vector, for all data samples,

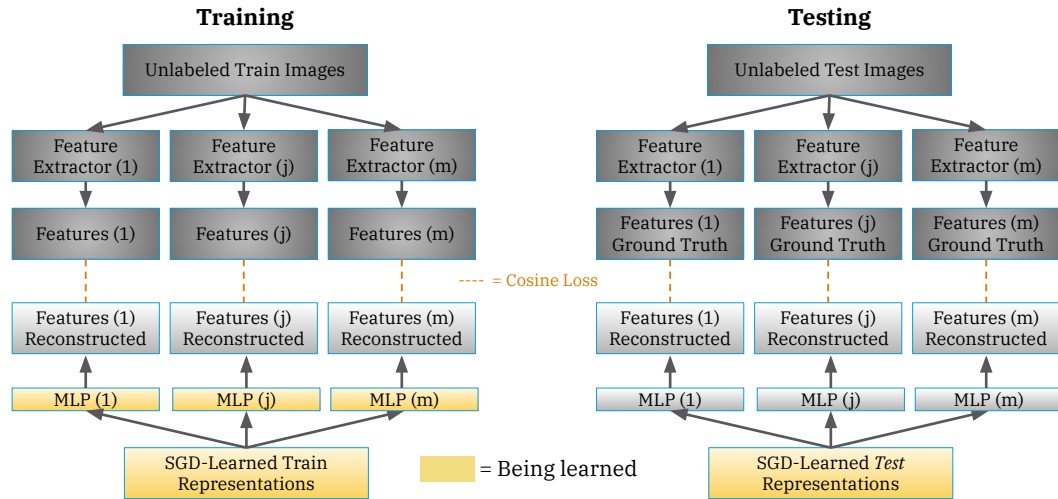

Figure 1: A schematic of our method. We wish to ensemble $M$ feature extractors over a training dataset of $n$ images, $M$ MLPs are initialized as well as representations for each of the $n$ images. The training objective is for all of the features of the $M$ models to be recoverable by feeding the learned representations through the respective MLP. The MLPs and learned representations are simultaneously optimized via gradient descent, using a cosine loss. At inference time, the MLPs are frozen, and solely the image representation is optimized.

---

**Algorithm 1** PyTorch-like pseudocode of our method

---

```
# Training Phase
# Initialize representations to average feature
train_feature_list = [net(images) for net in ensemble]
avg_feat = average(train_feature_list) # shape = (n_points x feature_dimension)
learned_train_reps = Parameter(avg_feat.detach()) # initialize params with avg_feat
mlps = [MLP() for net in ensemble] # 1 net per feature extractor
opt = SGD(mlps.parameters() + learned_train_reps) # optimize both mapping MLPs and input
    representations
for images_idx, images in trainloader:
    ensemble_feats = [net(images) for net in ensemble] # Get ensemble features
    outputs = [mlp(learned_train_reps[images_idx] for mlp in mlps] # Map learned
        representations through different MLPs
    loss = cosine_loss(ensemble_feats, outputs)
    loss.backward()
# Inference Phase
test_feature_list = [net(images) for net in ensemble]
learned_test_reps = average(test_feature_list)
opt = SGD(learned_test_reps) # Freeze MLPs at inference time
for images_idx, images in testloader:
    # Same as training loop
```

---

MLP(): a multi-layer perceptron model
Parameter(t): Pytorch function that takes the argument array t and initializes trainable parameters with that value

then such representations are useful. While concatenation of features can trivially achieve this object, we show that such an operation is in fact suboptimal in terms of the behavior of the derived feature space. We instead propose to *directly* learn via gradient descent a set of representations that contain all of the information necessary to derive the ensemble features. Our architecture is shown in Figure 1, with example pseudocode in Algorithm 1. Specifically, we show that extracting features from an ensemble of self-supervised models using this technique improves the nearest neighbor (NN) performance when evaluated on downstream supervised tasks.

## 2  RELATED WORK

**Supervised ensembling**: It is a ubiquitous technique in machine learning (Dietterich, 2000). In addition to the large number of online contests won through such approaches (Andres), ensembling has

also been demonstrated to achieve state-of-the-art performance on standard computer vision benchmarks (Huang et al., 2017). Most approaches employ Bayesian ensembling, where the predictions of separate networks are averaged (Wu et al., 2021; Hinton et al., 2015; Dietterich, 2000). While this relied on the alignment of objectives between the networks, we show that such an averaging on the intermediate features does indeed generate representations superior to that of individual models. Our method differs from this literature, however, in the learning done on top of the ensemble as well as the no-label setting of our representation learning.

**Knowledge distillation** (Hinton et al., 2015): This is a related vein of work is that of where the knowledge of an ensemble is distilled into a single neural network, or a single model is distilled into another network. This second version, known as self-distillation (Zhang et al., 2019), is highly relevant to our single-model setting, as we are able to obtain improvements while operating purely in the *feature* space of a single model. Our goal is not to discard the ensembled models as it is in knowledge distillation, but our method bears similarities to that of Hydra (Mandt et al.), where a network is trained to output representations capable of recovering the ensembled outputs. We note that our resulting accuracies consistently surpass average ensembling, a baseline that Hydra considers as an upper-bound to their method (but the differing settings do not lend themselves to apples-to-apples comparisons).

Xu et al. (2020) propose a self-supervised knowledge distillation algorithm that uses both supervised and self-supervised loss to train a teacher network, and then distill this combined knowledge to a student network. They show that this combination improves the student performance compared to traditional KD. In contrast to their work, our goal is not to learn a separate student network and we do not assume access to labeled data, but instead, our main goal is to extract rich nearest neighbor representations from an ensemble of self-supervised models.

$k$-**Nearest neighbor** ($k$-NN) **classifiers**: $k$-NN is a non-parametric classifier which has been shown to be a consistent approximator (Devroye et al. (1996)), i.e., asymptotically its empirical risk goes to zero as $k \to \infty$ and $k/N \to 0$, where $N$ is the number of training samples. While these theoretical conditions may not be true in practice, a main advantage of using $k$-NN is that it is parameter-free outside of the choice of $k$ (which typically does not change qualitative rankings) and thus is a consistent and easily replicable measurement of representation quality. Additionally, $k$-NN makes the decision process interpretable (Papernot & McDaniel, 2018; Dalitz, 2009), which is important in various applications (Mehrabi et al., 2021; Vellido, 2019; Gilpin et al., 2018). For these reasons, our paper focuses on extracting rich features from self-supervised ensembles that are condusive to $k$-NN classification.

On a related note, SLADE (Duan et al., 2021) leverages unlabeled data to improve distance metric learning, which is essentially the setting of our evaluation framework. While the goal is similar, SLADE uses supervised training to initialize learning and generate pseudolabels for the unlabeled points, our method assumes *zero* label access. Additionally, SLADE is concerned with learning a new network from scratch, as opposed to an ensembling framework.

**Pretrained Models:** Our method relies on the efficacy of pretrained self-supervised models. Specific methods we employ are SimCLR, SwAV, Barlow Twins, RotNet, PIRL, as well as traditional label supervision. In addition to the above works: Goyal et al. (2019); Zhai et al. (2019); Kolesnikov et al. (2019) benchmark and demonstrate the generalization efficacy of pretrained models in various settings.

**Gradient descent at inference time**: One atypical facet to our method is the use of gradient descent at inference time to directly learn new representations. While this approach is quite uncommon, we are not the first to leverage backpropagation at inference-time. Zadeh et al. (2019) uses backpropagation at inference time to learn representations from images using a generative "auto-decoder" framework and Park et al. (2019); Sitzmann et al. (2020) employ similar approaches to learn implicit representations of shapes. Sun et al. (2020) considers new samples as one-image self-supervised learning problems, and perform a brief optimization of a self-supervised learning objective on a new image (modifying the feature extractor's parameters) before performing inference, and Shocher et al. (2018) train a CNN at test-time for the purpose of super-resolution.

## 3 METHODS

We present a method for directly learning rich ensembled unsupervised representations of images through gradient descent. Consider a training collection of images $X = \{x_i\}_{i=1}^n$ and an ensemble of convolutional neural networks feature extractors $\Theta = \{\theta_j\}_{j=1}^m$. In this work the $\theta_j$ have previously been trained in a self-supervised manner on ImageNet classification and are ResNet-50s (Deng et al., 2009; He et al., 2016). Denote the L2-normalized features obtained by removing the linear/MLP heads of these networks and extracting intermediate features post-pooling (and ReLU) as $Z = \{\{z_i^{(j)}\}_{i=1}^n\}_{j=1}^m$, here $z_i^{(j)}$ denotes the intermediate feature corresponding to $\theta_j(x_i)$.

We initialize a memory bank of representations of $X$, with one entry for each $x_i$ these entries have the same feature dimensionality as the $z_i^j$. This memory bank is analogous to the type use in early contrastive learning works such as Wu et al. (2018). We denote this memory bank $\Psi = \{\psi_k\}_{k=1}^n$. Each $\psi_k$ is initialized to the L2-normalized average representation of the ensemble $\psi_k = \frac{\sum_{j=1}^m z_k^j}{||\sum_{j=1}^m z_k^j||}$, note that the sum operation is equivalent to averaging due to the normalization being performed.

To map the memory bank to the ensembled features, we employ a set of multi-layer perceptrons (MLPs), $\Phi = \{\phi_\ell\}_{\ell=1}^m$, each corresponding to a feature extractor $\theta_j$. Unless noted otherwise in our experiments, these $\phi_\ell$ are 2 layers, both of output dimension the same as their input (2048 for ResNet50 features). ReLU activations are used after *both* layers, the first as a traditional activation function, and the second to align the network in mapping to the post-relu set $Z$.

During training, a batch of images $\{x_i\}_{i \in I}$ are sampled with indices $I \subset \{1...n\}$. The corresponding ensemble features, $Z_I = \{\{z_i^{(j)}\}_{i \in I}\}_{j=1}^m$, are retrieved as are the memory bank representations $\Psi_I = \{\psi_k\}_{k \in I}$. Note that no image augmentations are included in our framework, meaning that the $z_i^{(j)}$ are typically cached to lessen computational complexity. Each banked representation is then fed through *each* of the $m$ MLPs, $\Phi$, resulting in a set of mapped representations $\Phi(\Psi_I) = \{\phi_\ell(\psi_i)\}_{\ell \in \{1...m\}, i \in I}$. The goal of the training is to maximize the alignment of these mapped features $\Phi(\Psi_I)$ with the original ensemble features $Z_I$. This is done by training both the networks $\Phi$ and the representations $\Psi$ using a cosine loss between $\Phi(\Psi_I)$ and $Z_I$, gradients are computed for both the MLPs and representations for each batch.

Once training is completed, the $\phi_\ell$ are frozen. During inference, when a new image $x'$ is given, the above process is repeated with the frozen MLPs. Concretely, the ensemble features $\phi_\ell(x')$ are computed and averaged to initialize a new representation $\psi'$. $\psi'$ is then optimized via gradient descent to maximize the cosine similarity of each $\phi_\ell(\psi')$ with $\theta_\ell(x')$, $\psi'$ then serves as our representation of $x'$.

The described method results in representations *superior to either the average or concatenated feature in terms of nearest-neighbor accuracy*. We highlight several exciting aspects of our method:

- The learning of a representation at test time via gradient descent is an uncommon approach. Existing methods do exist, such as auto-decoders in Zadeh et al. (2019) or implicit 3D representation literature (Park et al., 2019; Sitzmann et al., 2020). There are also techniques, such as Sun et al. (2020) for generalization, which use gradients to shape the *parameters* prior to inferenc.

- The vast majority of ensembling literature focuses on the *supervised* setting, where the training objectives of the ensembled networks are identical and thus aligned. Very little work has been performed on improving a group of self-supervised features with outside auxiliary signal. Hydra (Mandt et al.) considers a similar setting, but with a focus on knowledge distillation of the ensemble into a single network.

- Our method is extremely adaptable to different settings. Networks trained on multiple objectives, with different head architectures, can be usefully ensembled as demonstrated in Sec. 4. This could trivially be extended to using networks of different architectures as well (e.g. VGG + ResNet + AlexNet) (Simonyan & Zisserman, 2014; He et al., 2016; Krizhevsky et al., 2012). The flexibility of our approach additionally extends to the input data. While we use networks pretrained on ImageNet, the ensembling provides benefit on *both ImageNet as well in the self-supervised transfer learning setting*. This transfer can

either be performed by training new $\Phi$ on the target dataset or by training $\Phi$ on ImageNet and then using the frozen MLPs to learn representations on the target dataset.

- Our method as presented requires *only a single forward pass of each image through each ensembled model* as we do not use data augmentation. This allows caching of CNN features and fast training.

## 3.1 Technical Details

**Models**   The MLPs are as previously described. The pretrained ensemble $\Theta$ consists convolutional neural networks (all ResNet50s with features extracted between the stem and head of the network) that have had pretraining on ImageNet. The method used in said pretraining varies and is a subject of study in this work. Methods considered include SimCLR(v2), SwAV, Barlow Twins, PIRL, Learning by Rotation (RotNet, Gidaris et al. (2018)), and supervised classification. Pretrained models are obtained from the VISSL Model Zoo (Goyal et al., 2021), specific model choices are provided in the appendix.

**Data**   For our ImageNet experiments, we use the standard ILSVRC 2012 (Deng et al., 2009) training/validation split with per-channel normalization. We additionally include several datasets to evaluate the efficacy of our method in a self-supervised transfer learning setting. These datasets are CIFAR10/100, Street View House Numbers (SVHN), Food101, and EuroSat with splits as provided by VISSL, the same per-channel normalizations are used when inputting these images into the pretrained $\theta_j$ (Krizhevsky et al., 2009; Netzer et al., 2011; Bossard et al., 2014; Helber et al., 2019). No data augmentations are used in this work.

**Training**   Adam optimizers with a learning rate of $3 \cdot 10^{-4}$ are used throughout all experiments with batch sizes of 4096 for all ImageNet training and 256 on all other settings (Kingma & Ba, 2014). For ImageNet experiments, 20 epochs are performed for both training and inference, otherwise, 50 epochs are used. In practice, we find it useful to warmup the MLPs ($\Phi$) for half of the training epochs before allowing the learned representations to shift from their average initialization.

**Evaluation**   We employ k-nearest neighbor ($k$-NN) evaluation throughout to measure the quality of features, qualitative comparisons remain constant under variance in the choice of $k$. The choice of k-nearest neighbor is made as it is a parameter-free evaluation method that requires minimal tuning. Under regularization-free linear evaluation transfer learning, our ensembled features outperform both our ensembled feature baselines as well as their component state-of-the-art transfer learning models (e.g. SimCLR). Under heavy regularization cross-validation as standardized in self-supervised learning works such as Grill et al. (2020); Kornblith et al. (2019), however, the ensembled baselines can exceed performance of all of the above (see Appendix).

**Baselines**   We compare against several ensembling baselines throughout this work.

- *Concatenation*: for image $x_i$ the concatenation of all $z_i^{(j)}$ into a single vector $z_i^c \in \mathcal{R}^{2048m}$

- *Averaging*: the average feature $\frac{\sum_{j=1}^{m} z_k^j}{||\sum_{j=1}^{m} z_k^j||}$ (the initialization of our learned $\Psi$)

- *Individual*: a single model of the ensemble being used (in each case we detail specifically *which* model this is).

## 4 Results

### 4.1 Ensembling

In Figures 2 and 3, we use our method on an ensemble consisting of 4 SimCLR models. Figure 2 demonstrates the efficacy of our method on the *source* dataset, ImageNet. All of the ensembled models were trained in a self-supervised fashion on this source, but our method extracts an additional 2% of performance, increasing the nearest-neighbor accuracy to over 58%.

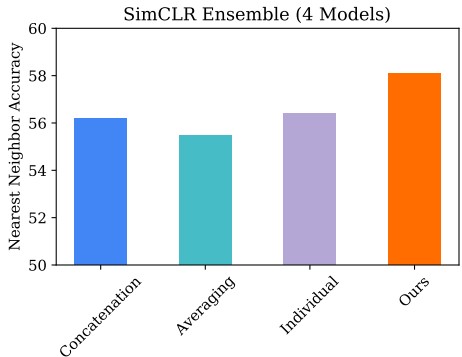 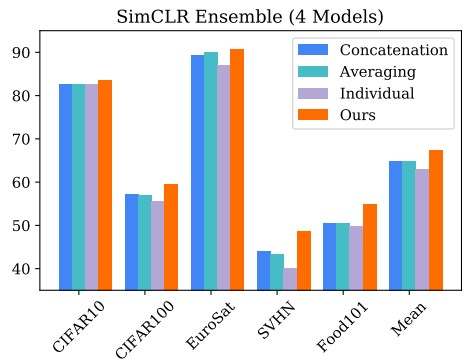

Figure 2: Nearest neighbor accuracies on the validation split of ImageNet. Our method improves over all baselines by over 2%.

Figure 3: Our method applied to non-ImageNet datasets, leveraging the generalization of our pretrained feature extractors. Performance is improved across all datasets.

In Figure 3, learning is performed on novel datasets, this can be thought of as self-supervised transfer learning. Labels are not made available until evaluation time, and then only to measure the $k$-NN accuracy. Using the frozen SimCLR features extractors, our model learns representations which achieve over 2.5% higher $k$-NN accuracy on average, with a smallest win of 0.6% (over Averaging on EuroSat).

We employ different types of ensemble in Figures 4 and 5. Figure 4 employs an ensemble consisting of five differently trained self-supervised models: Barlow Twins, PIRL, RotNet, SwAV, and SimCLR. These represent various approaches to self-supervised learning: SwAV and SimCLR are more standard contrastive methods, while Barlow Twins achieves state-of-the-art performance using an information redundancy reduction principal. SwAV is a clustering method in the vein of DeepCluster (Caron et al., 2018) and RotNet is a heuristic pretext from the family of Jigsaw or Colorization (Noroozi & Favaro, 2016; Zhang et al., 2016). Here we use Barlow Twins as the "Individual" comparison as it notably achieves the highest individual $k$-NN accuracy on every dataset. This setting is intriguing from two different perspectives: firstly, the varying strengths of the underlying ensembled models is challenging as noisy signal from the weaker models can drown out that of the strongest; second, the varied pretraining methods results in different strengths. For example, RotNet is by far the weakest model of the ensemble with an average transfer $k$-NN accuracy almost 10% lower than any other model. However, on SVHN (a digit recognition task), it performs very well, beating all non-Barlow methods by over 4% (the efficacy of such geometric heuristic tasks on symbolic datasets as previously as been noted in Wallace & Hariharan (2020)). Our model seems to benefit from this, achieving its largest win over Barlow Twins (8.2%) on this dataset. This demonstrates the ability of our model to effectively include multiple varying sources of information.

In Figure 5, we consider the effect of using our method on a *supervised* ensemble. While the pretraining goals of these models are aligned and thus traditional techniques (e.g. prediction averaging) could be used, we demonstrate that our model successfully improves upon the ensembled intermediate features further demonstrating its agnosticity towards pretraining tasks.

## 4.2 EFFICACY ON INDIVIDUAL MODELS

While our method is designed as an ensembling technique, we discover that it is surprisingly effective when employed on *a single model*. These result are quite remarkable, as the improvement of features without access to their corresponding images or additional supervision is quite challenging. This setting is especially remarkable as here *the input initialization and targets are identical*; we find that the MLP, $\phi$, does not converge to a perfect identity function during the warmup period and the movement of the representations $\psi$ in fact help enable near-perfect target recovery. In the inference stage, the MLP output of the average feature is close to identity (0.97 cosine similarity), but only by learning the representation through gradient descent does the similarity improve to near-perfect (0.99+) similarity. We discuss possible reasons for this behavior in Section 5.

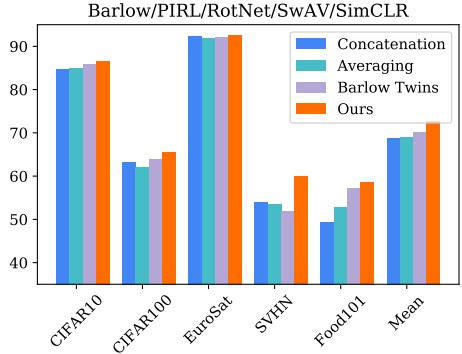

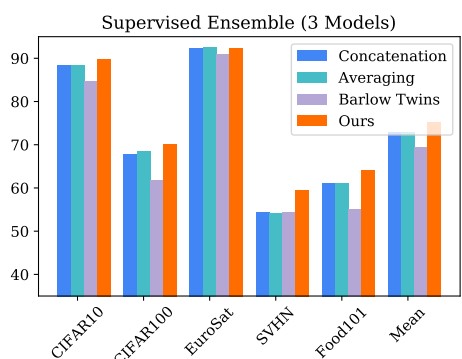

Figure 4: Ensembling varied models. We observe that our method seems to capture specialties/strengths of the component feature extractors, particularly the symbolic-dataset efficacy of RotNet.

Figure 5: Despite not utilizing the consistency of the supervised classification objective, our method effectively combines supervised models to improve upon the performance of all other ensembles considered.

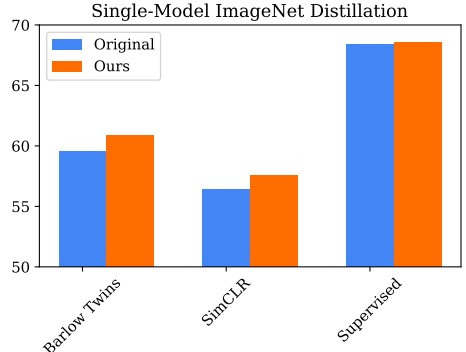

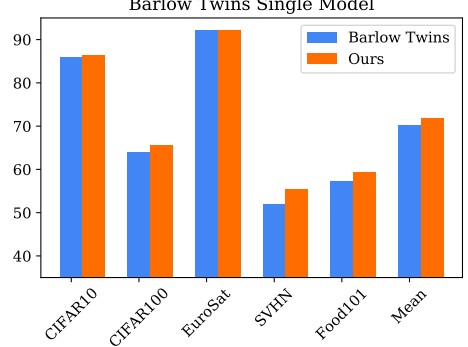

Figure 6: Paralleling the efficacy of self-distillation (Zhang et al., 2019), our "ensembling" method proves to provide performance gains even when just one model is employed.

Figure 7: We experiment with a single Barlow Twins model on our generalization benchmark. Performance is gained on all datasets, with a mean improvement of over a full percent.

In Figure 6, we see that our "ensembling" technique benefits all individual models substantially (1.8, 1.3 and 0.4% respectively) when our representations are trained on ImageNet. Given the degree to which the original self-supervised models' objectives are optimized, the margin of the improvement is quite impressive. This benefit carries over the self-supervised transfer learning as well (Figure 7). Here we use our method in conjunction with a Barlow Twins model. Our method offers a mean $k$-NN accuracy gain of over 1%, once again *despite no additional information, augmentations, or images being made available besides the CNN's features themselves*. While this gain is relatively minor, we show in the appendix that superiority to the baseline features is maintained across a wide range of hyperparameter choices.

## 4.3 TRANSFERRING MLPS FROM IMAGENET

So far we have only considered the scenario where the MLPs, $\Phi$, are trained on the same dataset as the representations $\Psi$, where inference is ultimately performed. This is not a necessary assumption of our framework, however, once MLPs are trained on a dataset they can be re-used to learn representations $\Psi$ on arbitrary imagery. We conduct experiments where $\Phi$ is trained on ImageNet (specifically those generated for use in Figures 2 & 6) are re-used in the *transfer* setting. Because the MLPs are frozen, no parameters of any networks are being changed during training, solely the representations $\Psi$ are being learned. The results of these experiments are shown in Figures 8 & 9.

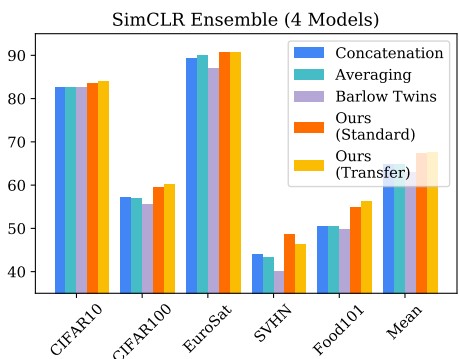 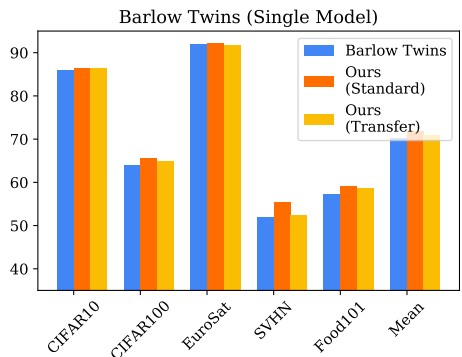

Figure 8: The MLPs do not necessarily need to be trained on the target dataset, but can be transferred from ImageNet to other downstream datasets. Here no parameter tuning is done on each dataset, gradients are computed solely to directly learn the representations. We find in fact that this transfer approach maintains the performance of directly learning new MLPs.

Figure 9: In the single-model case, transferring $\phi$ still provides benefit over the baseline, but is less effective than learning the MLPs per-dataset.

In the ensemble setting, the performance is largely maintained when re-using MLPs from ImageNet. Both method (*Ours (Standard)* and *Our (Transfer MLPs)*) are still superior to the baselines across *all* datasets. This result is intriguing, as the observed ensemble gains are relatively large, and this method requires *no* pretraining on a targeted transfer set, simply inference via gradient descent of $\Psi$; such generalization properties are a part of what makes the base pretrained models so valuable in application.

In the single-model setting, the Barlow Twins model + MLP trained on ImageNet is re-used across transfer datasets. Here we see a notable decline in performance, but the transferred model still maintains improvement over the baseline on 4 out of 5 datasets (all but EuroSat).

# 5 ANALYSIS

## 5.1 DEEPER $\Phi$ REGULARIZE $\Psi$

One hypothesis for the efficacy of our method in the single-model setting is that $\phi$ acts as a regularizer. The findings of Arora et al. (2019) demonstrate that stochastic gradient descent in deep neural networks tends to recover low-rank solutions when performing matrix factorization, this helps to explain the generalization properties of deeper networks: deeper networks lead to simpler solutions which tend to generalize better. This relates to our method, as our improvements hinge upon the network $\phi$ *not* learning a perfect identity function during warmup; if it did, then the gradients with respect to $\Psi$ would vanish and no change of representation would occur. Arora et al. (2019) suggests then that a deeper network might improve the ultimate quality of $\Psi$, as the points would need to be recoverable by a lower-rank MLP.

We confirm this phenomenon in Figure 10 by varying the depth of $\Phi$ from 1 to 8 layers while learning representations directly on our varied dataset benchmark using a Barlow Twins model. Increasing depth improves accuracy incrementally over a total of '% on average until the network is 6 layers deep, more than triple that of our default setting. Some but not all of this performance boost is recoverable by adding in small amounts of traditional weight decay (e.g. $1e-6$) to the parameters of the MLP.

## 5.2 BEHAVIOR OF MODEL

Now that we have established a partial explanation of why our model works, by learning representations which preserve information under the regularization of an SGD-learned deep network, we now investigate what specific changes our method makes to the feature space.

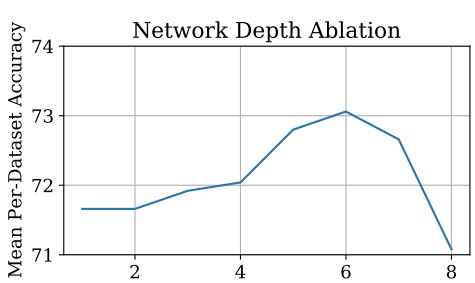

Figure 10: Ablation of MLP depth, possibly suggesting that the low-rank tendency of deeper networks serves as a regularizer on the learned representations. This results in improved representation quality with network depth up to 6 layers.

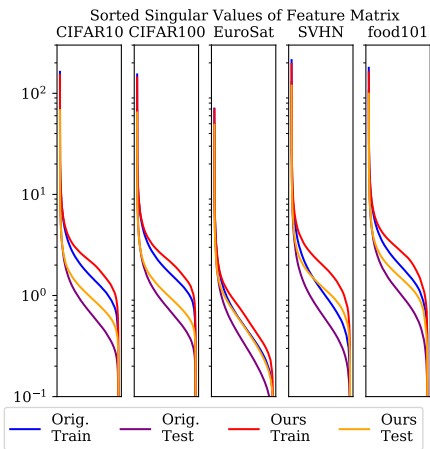

Figure 11: Sorted singular value curves for our method vs. the baseline features in an apples-to-apples setting (learning $\phi$ restricted to non-negative). Our method learns features with a more balanced set of singular values, indicating a more uniformly spread bounding space.

First, in Figure 11, we examine the distribution of $\Psi$. We do so by training representations from a single Barlow Twins model similar to previously, with the important distinction that we restrict our points to be non-negative (i.e. in the first n-tant of feature space), to make an apples-to-apples comparison to the baseline features. We examine the singular values of this (constrained) feature matrix compared to that of the original features. In general, the singular value distribution of $\Psi$ are less heavy-tailed: meaning the volume occupied by the features is larger and more uniform in each dimension than the baseline features. This is an indication of $\Psi$ learning a regularized form of the original $Z$. The above finding indicates that the feature representations are spread out as a result of the learning process. We also investigate what happens to clusters in Figure 15 in the Appendix. In summary, the findings suggest that our methods success is partially attributable to accentuation of existing clusters in the dataset.

## 6 Discussion

In this work, we presented a novel self-supervised ensembling framework which learns representations directly through gradient descent. The intuition behind our method is to capture all of the knowledge contained in the ensembled features by learning a set of representations from which the former are fully recoverable. We demonstrated the efficacy of our method in Section 4 and analyzed causes and effects of the representation improvement in Section 5. We hope that this work lays the groundwork for further forays into the problem of utilizing combinations of the powerful pretrained models that are becoming plentiful in the computer vision literature.

As previously noted, while we demonstrate improvement under $k$-NN in this paper the average feature baseline surpasses our method under linear evaluation when regularization is heavily optimized under a grid sweep as standardized in Grill et al. (2020); Kornblith et al. (2019). We tried applying various regularizations during MLP training, including traditional L2 weight decay, L1 regularization of $\Psi$, the dimensionality of $\Psi$, and the depth/width of the MLPs $\Phi$. While some of these modifications further increased the $k$-NN performance improvements, when representations were evaluated under linear regression with cross-validated regularization none consistently surpassed the average ensemble baseline.

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

## A APPENDIX

### A.1 ROBUSTNESS TO HYPERPARAMETER CHOICES

In Figures 12 and 13 we examine the robustness of our method (Single Barlow Twins Model on varied dataset benchmark) to choices of learning rate and batch size. We find that superior performance to the baseline is maintained across a wide choice of settings (despite this being one of the settings where our margin of improvement is the smallest) and that there is in fact room for further per-task optimization via cross-validation of hyperparameters.

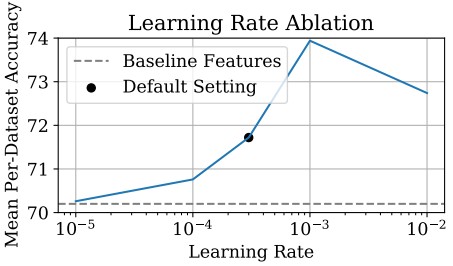

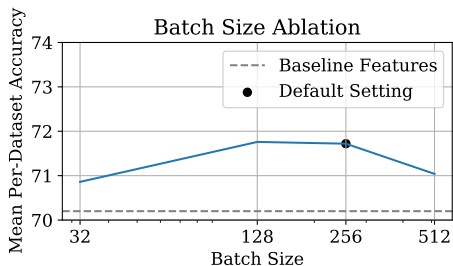

Figure 12: Mean per-dataset accuracy, single Barlow Twins model on varied dataset benchmark. Learning rate ablation.

Figure 13: Mean per-dataset accuracy, single Barlow Twins model on varied dataset benchmark. Batch size ablation. Linear learning rate scaling rule followed.

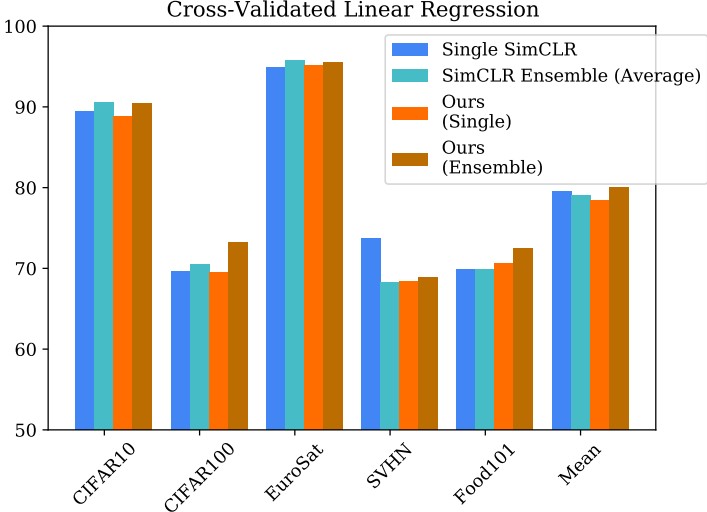

Figure 14: Linear regression accuracies with cross-validated L2-regularization. Relative performance of our method is worse compared to the $k$-NN evaluation setting.

## A.2 CROSS-VALIDATED LINEAR REGRESSION

We follow a protocol similar to Grill et al. (2020); Kornblith et al. (2019). A validation subset (10% of the training set) is sampled and held out during training. Training on the remaining 90% of the data is performed for a hyperparameter sweep with performance on the validation subset being measured. We employ an SGD optimizer for 1000 epochs with a batch size 4096 and learning rate 1.6. A weight decay sweep of $\lambda \in \{1e-6, 1e-5, 1e-4, 1e-3, 1e-2\}$ is performed. Results are shown in Figure 14. We see that our method in the single-model setting suffers substantially compared to the baseline. In the ensembled setting, there is still improvement on-average, but the gains are much more inconsistent than under $k$-NN evaluation. While this is a current limitation of our model the benefits under $k$-NN indicate a fundamental utility in our ensembling method to learn new reprsentations.

## A.3 MODELS USED

**Barlow Twins**

- https://dl.fbaipublicfiles.com/vissl/model_zoo/barlow_twins/
  barlow_twins_32gpus_4node_imagenet1k_1000ep_resnet50.torch

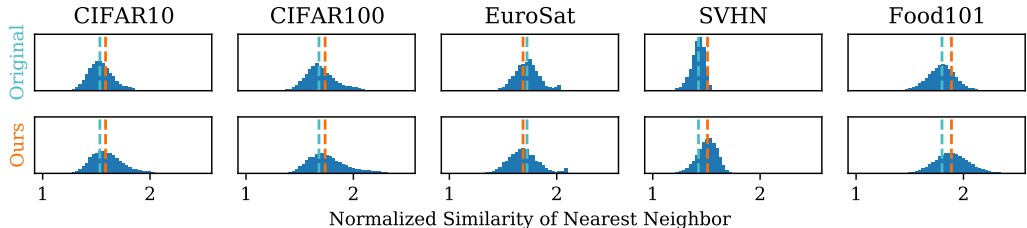

Figure 15: The normalized maximum similarity (measure of how close each test point's nearest neighbor in the trainset is relative to average) for each method-dataset pair. Dashed lines indicate the median for each distribution We see that the proposed approach generally has higher normalized maximum similarities, indicating relatively tighter clustering behavior.

**SimCLR**

- 200 epoch training: `https://dl.fbaipublicfiles.com/vissl/model_zoo/simclr_rn50_200ep_simclr_8node_resnet_16_07_20.a816c0ef/model_final_checkpoint_phase199.torch`

- 400 epoch training: `https://dl.fbaipublicfiles.com/vissl/model_zoo/simclr_rn50_400ep_simclr_8node_resnet_16_07_20.36b338ef/model_final_checkpoint_phase399.torch`

- 800 epoch training: `https://dl.fbaipublicfiles.com/vissl/model_zoo/simclr_rn50_800ep_simclr_8node_resnet_16_07_20.7e8feed1/model_final_checkpoint_phase799.torch`

- 1000 epoch training: `https://dl.fbaipublicfiles.com/vissl/model_zoo/simclr_rn50w2_1000ep_simclr_8node_resnet_16_07_20.e1e3bbf0/model_final_checkpoint_phase999.torch`

**Supervised**

- `https://download.pytorch.org/models/resnet50-19c8e357.pth`

- `https://dl.fbaipublicfiles.com/vissl/model_zoo/converted_vissl_rn50_supervised_in1k_caffe2.torch`

- `https://dl.fbaipublicfiles.com/vissl/model_zoo/sup_rn50_in1k_ep105_supervised_8gpu_resnet_17_07_20.733dbdee/model_final_checkpoint_phase208.torch`

**Varied Models**

- PIRL: `https://dl.fbaipublicfiles.com/vissl/model_zoo/pirl_jigsaw_4node_pirl_jigsaw_4node_resnet_22_07_20.34377f59/model_final_checkpoint_phase799.torch`

- RotNet (note, trained on ImageNet-22k): `https://dl.fbaipublicfiles.com/vissl/model_zoo/converted_vissl_rn50_rotnet_in22k_ep105.torch`

- Swav: `https://dl.fbaipublicfiles.com/vissl/model_zoo/swav_in1k_rn50_800ep_swav_8node_resnet_27_07_20.a0a6b676/model_final_checkpoint_phase799.torch`

## B BEHAVIOUR OF MODEL (ADDITIONAL EXPERIMENTS)

This is a continuation of section 5.2 in the main text.

Here we wish to investigate what happens to clusters. We examine this again through the lens of nearest-neighbors: for each $\psi'_k$ in the test set, we calculate the maximum cosine similarity of points in the train set. This maximum similarity is then normalized by the *mean* similarity for each method-dataset pair. The resulting normalized similarity provides a measurement of how relatively close points are to their nearest neighbor vs. an average pair of points (with a higher value indicating relative closeness). Histograms of this metric are shown in Figure 15. For 4 out of the 5 datasets, our method results in tighter neighbor matchings than the baseline. Intriguingly, the one dataset for which this does not hold is EuroSat, is also the dataset where our models consistently yielded the lowest benefit. These findings suggest that our methods success is partially attributable to accentuation of existing clusters in the dataset. It is interesting to note as an aside that the mean pairwise similarity across the entire dataset is significantly lower for our method, cementing the findings from Figure 11 that the features are more distributed across space.

