# OpenReview forum: "Learning Rich Nearest Neighbor Representations from Self-supervised Ensembles"
_ICLR.cc/2022/Conference — ICLR 2022 Submitted_

### Official Review · Reviewer_HHFN · 2021-10-23

**Correctness:** 3
**Technical Novelty And Significance:** 2
**Empirical Novelty And Significance:** 4
**Recommendation:** 5
**Confidence:** 4

**Main Review:**

**Strengths**
1. Figure 1 and Algorithm 1 do a good job in explaining the proposed method.
2. I appreciate the extensive experiments the authors have done both on ImageNet and on other datasets.
3. I appreciate the authors specify the checkpoints used in the experiment and clarify all the hyper-parameters.

**Weaknesses**
1. Paper writing.
- The description of the method in general needs to be improved. The figure 1 and algorithm 1 helps a lot with understanding the proposed method, however, the description itself in Section 3 is not clear. I find it difficult to understand which parameters are optimized during the training, it takes me quite a while to get that the inputs to the MLPs are optimized, but not the backbone parameters, this may leads to some confusion in understanding the proposed method.
- The description of the some key components of the method is not clear. For example, the details of the kNN evaluation. At the beginning of section 3, the authors only mentioned which feature is used for the test images. However, they do not specify which image feature is used for the training images, this makes the kNN method in the evaluation in section 3.1 not clear. There are a variety of choices for training images features that I can think of to be used in the kNN. For example, this could be the $z_i^j$s, $\psi_k$s, or $\phi_{l}(\psi_k)$s. I think this issue needs to be addressed since which feature is used in the kNN may make a huge difference in terms of the experiment result.

2. I do not get the idea behind the proposed inference procedure.
- For example, why frozen MLP? What happens if the MLPs are not frozen?
- Why use $\psi'$, what happens if $\bar{\theta}_{\ell}(x')$ is optimized and used for the inference.

3. More experiments need to be done in order to show the superiority of the proposed method. Below are several experiment that the reviewer is interested.
- What if different training and/or test feature are used in the kNN, would this make a difference in terms of the experiment result?
- The image feature is optimized by maximizing the cosine similarity in the proposed approach, comparing to the baseline methods used for comparison in the experiments. What if other self-supervised learning approaches used, would the proposed method still be useful? For example, what if minimize the L2 distance between the reconstructed feature and the extracted feature.
- Based on the experiment result in Section 4.3, it seems like the parameter values of the MLPs does not matter, but optimizing the representation matters. A further experiment should be carried out to see what happens if random initializations for the MLPs are used and are fixed.
- What happens if MLPs are not frozen during the test?
- In Figure 15, the authors find the clusters produced by the proposed method has larger within class similarity and may lead to the better performance. If this is the case, another baseline for comparison should be the $z_j^i$s are all included in the kNN. In another word, there are m*n test image features.

**Minor Issue**
- The paper needs more careful proofread. For example, the legends in the figures are out of the box (see figure 14), the sentence in caption of Figure 15 is not complete. The citation of Mandt et al. is not right.

**Summary Of The Paper:**

The paper proposes a new self-supervised ensembling method to get a better feature representation. Instead of using the conventional averaging or concatenating to ensemble multiple features, this paper proposes a different inference scheme where the backbone is also updated during the test time in a self supervised fashion. Experiments are conducted to empirically show the superiority of their methods compared to some existing methods.

**Summary Of The Review:**

The overall approach of using self-supervised ensembling may potentially be interesting for the community. However, some details of the proposed method needs to be addressed and more experiments need to be done to explore potential weakness/strengths of the proposed method. I am happy to review the rating if these concerns are addressed.

---

### Official Review · Reviewer_TyZ7 · 2021-11-02

**Correctness:** 3
**Technical Novelty And Significance:** 2
**Empirical Novelty And Significance:** Not applicable
**Recommendation:** 3
**Confidence:** 4

**Main Review:**

strength:
The paper is well-written clearly structured in general. And the performance looks well on the K-NN classification.

weakness:
The paper proposes a simple ensembling method that optimizes the self-supervised pretrained model. But some results and explanations are not very clear in the paper and the experimental section has to be improved. I think the contribution of this paper is not significant enough for ICLR.

**Summary Of The Paper:**

This paper propose a new way to learn self-supervised model ensembling. Their novel approach learns representations via gradient descent directly at inference time after having pretrained feature extractors. And the authors conduct a series of experiments to show the efficacy of their method.

**Summary Of The Review:**

1) As mentioned in the appendix, the ensembling method seemingly does not work on the linear regression evaluation but only boosts the performance on the K-NN classification. How to explain that phenomenon? Why not use the linear probing or finetune protocol to evaluate the representation? Also, I suspect the superior performance is highly dependent on the self-supervised methods, e.g., Barlow Twins，MoCo, SimcCLR, etc. Since the feature extractor is the ultimate goal in self-supervised learning, it would be more convincing if authors can propose an approach to train end-to-end in a self-supervised manner.

2) Does the proposed method only boost K-NN classification performance? If so, I think the contribution of this work is not enough because K-NN classification is only a very limited downstream task for self-supervised learning. In addition, the proposed method need to be trained based on a pretrained model, and be fine-tuned during test stage. Compared with the cost of time, the improvement is trivial.

---

### Official Review · Reviewer_94Ck · 2021-11-02

**Correctness:** 2
**Technical Novelty And Significance:** 2
**Empirical Novelty And Significance:** 2
**Recommendation:** 5
**Confidence:** 3

**Main Review:**

1. The improvement is marginal considering the training and testing cost (20 epochs on ImageNet and 50 epochs on other datasets). Noting that the proposed method needs to fine-tune during validation. Is there any possibility of information leakage from validation data?
2. This paper only compares the proposed method with the provided baseline. It is necessary to compare with other advanced ensembling methods.
3. What are the differences between "4 SimCLR models" in Fig. 2?
4. Some experimental issues:
  - The baseline performance seems to be too low in Fig. 2.
  - Only KNN results are reported. What about the classification accuracy under the linear evaluation protocol, which is widely adopted in self/un-supervised methods?
  - It is claimed that no data augmentation is used. What about the resizing augmentation

**Summary Of The Paper:**

This paper proposes a framework to perform self-supervised model ensembling via a novel method of learning representations directly through gradient descent at inference time. The effectiveness of the proposed method is evaluated by k-nearest neighbors accuracy.

**Summary Of The Review:**

The main concern comes from the fine-tune manner in the testing phase, which may cause cheating for the model from the validation data. The key contribution needs to be further clarified as the applicable scenario for this method seems to be limited.

---

### Official Review · Reviewer_mwTd · 2021-11-03

**Correctness:** 3
**Technical Novelty And Significance:** 2
**Empirical Novelty And Significance:** 2
**Recommendation:** 5
**Confidence:** 4

**Main Review:**

Pros
1) The problem of model ensemble for self-supervised learning studied in this paper is interesting and important. It could potentially improve the performance of current self-supervised learning methods and achieve robust results in real-world applications.
2) The proposed method of learning both feature vectors and MLP is reasonable.
3) Experimental results show that the proposed method can improve the performance of an existing representation from a pre-trained self-supervised learning model.

Cons
1) The main experimental comparisons are based on KNN classification without data augmentation, which is not a common practice in self-supervised learning. Self-supervised representation learning methods usually compare the quality of the learned representation on different tasks, for example linear classification, object detection or semi-supervised learning. The reported KNN result of a 58% accuracy is far behind the state-of-the-art performance of self-supervised learning on ImageNet.
2) I have doubts about the comparison with the learned representation vectors with baselines. As reported in Figure 7, the learning process with MLP has already improved upon original individual features with one model. Then it is unclear to me whether the method would still improve upon similar learnings on top of concatenation or averaging, since the baselines from Fig. 2 to 5 directly use features without learning if I am understanding correctly. Besides, I think an important baseline is missing, which is simply doing ensemble based on the outputs of linear classifiers trained on top of every models.
3) Figures in this paper are indistinguishable when not viewing in colour.

I am curious about the results in Figure 9. It seems that transferring the MLP does not lead to improved performance. But how many data examples are needed for the target task so that the MLP per-dataset can be well-trained to have the ability of proper generalization?

**Summary Of The Paper:**

This paper presents a representation learning method to optimize individual data representation vectors as well as an MLP encoder so that the learned representation vectors can recover the features from multiple pre-trained self-supervised learning models. Experimental studies show that such learning method can outperform baselines including individual feature, concatenation and averaging in terms of KNN accuracies.

**Summary Of The Review:**

This paper works on combining the pre-trained models from self-supervised learning, and the proposed method is well-motivated and reasonable, though not technically novel enough. However, experimental evaluations are not convincing enough and lacks comparisons in many situations.

---

### Decision · Program_Chairs · 2022-01-20

**Decision:**

Reject

**Comment:**

The paper proposes a method to perform self-supervised model ensembling by learning representations directly through gradient descent at inference. The effectiveness is evaluated by k-nearest neighbors accuracy.

The reviewers agreed that the paper studies an important and interesting problem of leveraging model ensembling for self-supervised learning, which could improve both the performance and robustness of the learned representations. However, the reviewers also agreed that there were issues with the soundness of the empirical evaluation, which was a key reason for rejection.